# Psychosocial Factors as Mediator to Food Security Status and Academic Performance among University Students

**DOI:** 10.3390/ijerph19095535

**Published:** 2022-05-03

**Authors:** Nor Syaza Sofiah Ahmad, Norhasmah Sulaiman, Mohamad Fazli Sabri

**Affiliations:** 1Department of Nutrition, Faculty of Medicine and Health Sciences, Universiti Putra Malaysia, Serdang 43400, Selangor, Malaysia; fiqzatuanahmad@gmail.com; 2Research Centre of Excellence for Nutrition and Non-Communicable Diseases, Faculty of Medicine and Health Sciences, Universiti Putra Malaysia, Serdang 43400, Selangor, Malaysia; 3Department of Resource Management and Consumer Studies, Faculty of Human Ecology, Universiti Putra Malaysia, Serdang 43400, Selangor, Malaysia; fazli@upm.edu.my

**Keywords:** university students, food insecurity, academic performance, anxiety, depression, mediator

## Abstract

The interrelation between food security, academic performance, and psychosocial factors remains unclear. This study aims to identify psychosocial factors as mediators of food security status and academic performance among university students at one of Malaysia’s public universities. Respondents included 663 bachelor’s degree students from seven randomly selected programmes at the university. Data on demographic and socioeconomic characteristics, food security status (US Adult Food Security Survey Module, FSSM), psychosocial factors (DASS-21), and academic performance were collected using an online survey. The SPSS PROCESS macro was used to perform mediation analysis. The result (β = −0.0182, *p* < 0.001) indicates that food insecurity is associated with poor academic performance. As well as that, an increase in anxiety (β = −0.0027, *p* < 0.05) and depression (β = −0.0025, *p* < 0.05) was significant associated with a decrease in academic performance. Furthermore, anxiety and depression serve as significant mediators in the relationship between food security and academic performance. Alleviating food insecurity is not only a way to improve academic performance; it can also improve academic performance by reducing anxiety and depression.

## 1. Introduction

Since the mid-twentieth century, there has been widespread interest about food insecurity worldwide, primarily in developing countries. Over the last thirty years, people’s perceptions of food security have shifted [1]. Food security is defined by the Food Agriculture Organization (FAO) (2009) as “when all people, at all times, have physical and economic access to sufficient, safe and nutritious food to meet their dietary needs and food preferences for an active and healthy lifestyle” [2]. This is the widely accepted definition used by researchers, which includes food access, availability, food utilisation and stability, and awareness of the importance of food to health [1].

Meanwhile, the United States Department of Agriculture (USDA) defines food security status as a range of levels, including high, marginal, low, and very low food security status. High and marginal food security are classified as food secure groups, whereas low and very low food security are categorised as food insecure groups. People who are food insecure have the following characteristics: lowered calorie intake, diets with a lack of diversity, starvation, not having access to healthy food, and reduced weight due to not having enough food. In addition, despite consuming enough calories, they may still be food insecure if the calories they consume are nutrient deficient [3].

Recently, there have been several studies about food security that focus on university students [4,5,6,7,8,9]. This issue is the hidden crisis among university students nowadays because they are not getting enough food to continue studying [10]. Students’ independence at university sharpens their decision making in all aspects of their lives, including their food preferences. Whatever they consume, it all depends on them: either they eat healthily or otherwise. Most studies on food insecurity in university/college students found that the results showed an alarming public concern where the high prevalence of food insecurity among students was at a very critical level, and this piqued the interest of other researchers [6,7,8,9].

Globally, many countries are experiencing food insecurity as it is an international issue that not only burdens developing countries but also developed countries such as the United States. Among studies in the United States’ colleges and universities, the average food insecurity among students is 32.9%, with a range between 14.5% and 58.8% at some point during their university life [10,11,12]. Food insecurity among students in the United States is even higher than the national average of 12.7% [13]. Prevalence studies that assessed the rate of food insecurity among university students have resulted in varying rates of food insecurity. 

In Malaysia’s context, at present, the high prevalence of food insecurity among university students has become an alarming issue for the public. According to a recent systematic review on food security in Malaysia, the prevalence of food insecurity among university students ranged from 22% to 70% [9]. According to Sulaiman et al., university students in Malaysia are struggling with their finances due to inadequate financial loans/scholarships and high living expenses [9]. In addition, a lack of financial management can lead to them not having enough money to buy food. Unplanned budgeting can lead to students’ losses; therefore, this leaves them prone to making unhealthy food choices.

University students are a special group of people who go through one of the most stressful events and critical transitions from adolescence to adulthood. Because of their importance in society and the country’s future, several international studies have found that food insecurity has a negative impact on students’ academic performance [10,14] as well as psychosocial factors such as stress, anxiety, and depression [15,16]. 

### 1.1. Food Security and Academic Performance

Food insecurity is the economic and social condition that interferes with students’ academic performance. The fundamental cause of food insecurity happening among students is that they do not have enough money to buy for their basic needs, which is food. Food insecurity has been associated with substandard academic performance [7]. Students are unable to consume a variety of nutritious foods as they usually cost more, leaving the body with insufficient nutrients. A lack of nutrition interferes with the learning process, probably because various nutrients are known to have essentials roles in brain development and functioning such as omega-3, folate, and iron. Moreover, constant energy is required for the brain to operate well [17]. In contrast, inadequate energy due to poor diet quality results in various health impacts, for example, low concentration in class, and impairs cognitive function that diminishes academic achievement [18]. Several researchers across the globe have discussed how food insecurity negatively affects university students’ academic performance [7,10,11,19].

According to Wattick et al., poor academic performance among food insecurity issues faced by students could be explained by several factors such as nutrient deficiencies, behavioural and psychosocial factors, or childhood academic impairment [18]. When students do not receive adequate nutrients that are essential for the development of the brain and growth during childhood, it leads to academic impairment and health problems. Children may become malnourished, and it also affects the behavioural and psychosocial well-being of the students where social cognitive impairment inhibits social interactions among peers [18]. Food insecurity causes students to develop low self-esteem and be unhappy because they are always hungry, which can lead to stress, anxiety, and depression in the long run. While numerous studies have examined how food insecurity is related to poor academic performance around the world, little is known about it in Malaysia, so this study took advantage of the opportunity to discuss food insecurity and poor academic performance among Malaysian students. Food insecurity is inadequate food in quantity and/or quality. The recent literature in Malaysia among university students suggested that food quality is positively associated with academic performance. With financial constraints, students tend to purchase unhealthy food over quality food and thus become fatigued, resulting in decreased concentration and academic performance [20]. 

### 1.2. Psychosocial Factors as Mediator

Though the series of studies explored the relationship between food insecurity and poor performance academically, the underlying mechanism remains blurred. As above mentioned, not having enough money to buy food can cause a person to be very stressed, anxious, and depressed every time they want to purchase meals. Inadequate nutrition interferes with students’ ability to focus on learning, thus impairing academic performance. Psychosocial factors are the characteristics that influence a person’s physical and social well-being [21]. 

The term “psychosocial” is very well known and is commonly used to describe the phenomenon where the outcome of social impacts is sometimes intermediated through psychological makeup. Psychosocial factors consist of protective psychosocial resources as well as psychological risk factors. Psychosocial resources include social networking and social support. To cope with these psychosocial resources, coping ability, self-esteem, and a sense of coherence are examples of factors needed. Meanwhile, the psychological risk factors take in depressiveness, hostility, and exhaustion [22]. 

Psychosocial factors such as stress, anxiety, and depression often influence the health outcome and performance of the general population [23]. Stress, anxiety, and depression are different in terms of severity but are interrelated and affect one’s psychosocial well-being. Stress is the body’s response to pressure in daily activities or life. Stress helps students accomplish goals and finish work. However, too much stress heightens anxiety and depression. On the other hand, anxiety usually lasts longer compared to stress. Feeling excessively anxious can build up an anxiety disorder, and those with long-term anxiety often encounter episodes of low mood and feeling hopeless (signs of depression) [24]. One can have both anxiety and depression as they share similar symptoms, and a lot of people with depression have experienced anxiety disorders previously in life [25]. Nevertheless, stress, anxiety, and depression affect students’ lives and action needs to be taken if they have symptoms [26].

The relationship between food insecurity and psychosocial factors has been thoroughly discussed in the literature [7,27,28]. Having food insecurity may add additional stress to an individual, contributing to poor mental health and, hence, not providing the brain enough nutrients to function well [18]. On the other hand, food insecurity can really give students a hard time keeping track of how to feed themselves as well as struggling to study well [27]. In addition, a previous study found that food insecure freshmen students had a higher chance of experiencing depression than food secure students. The researcher concluded that food insecurity is associated with health outcomes among university students [10]. In a similar study of undergraduate students, researchers discovered that students who were food secure were less likely to report depressive symptoms than food insecure students [14]. 

In contrast, multiple studies have reported that depression, anxiety, and stress are harmful for university students academically as they are prone to this problem due to various psychosocial changes, coping with the learning process and social demand, as well as future planning [29]. Properly managing and planning their own daily schedule is certainly one of the stressful events encountered by students; therefore, students who experience stress perform poor academically [29,30]. On the other hand, depression and anxiety have also shown similar results as stress [15]. Feeling depressed and anxious often impact students’ memory and concentration due to low mood, feeling hopeless, a lack of energy, reduced cognitive function, and insufficient sense of coping. These will result in difficulty gaining new knowledge and dealing with tests [31].

Indeed, previous studies have proven that mental health mediates the relationship between food security status and academic performance among students [7,8]. In line with previous research, this study attempts to focus on students from one public university in Malaysia. 

### 1.3. Present Study

Excellent academic performance helps students with job opportunities, stability, and helps the future nation. Food insecurity is often associated with higher odds of emotional distress (stress, anxiety, and depression) as well as hindering academic performance. It is important to identify the potential pathway to help in promoting quality of life for students. Early intervention on tackling a food insecurity issue can help reduce the emotional distress encountered by a student, thus improving performance academically. On the other hand, reducing stress, anxiety, and depression can also promote better achievement. Therefore, understanding the potential mediating variables can provide a better insight into the underlying pathway on the relationship between food security status and academic performance for better future intervention planning.

The present study seeks to address this gap by considering mechanisms that can potentially mediate the relationship between food security status and academic performance among university students in Malaysia. The interrelation between food security status, psychosocial factors, and academic performance remains unclear. Exploring this relationship can help future researchers have a better understanding of the underlying mechanism. To date, there has not been a study conducted in Malaysia that tested the mediating effect of psychosocial factors on food security status and academic performance among university students. Therefore, this study attempts to determine the potential mediators in the relationship between food security status and academic performance through psychosocial factors (stress, anxiety, and depression) among university students in Malaysia. 

## 2. Materials and Methods

### 2.1. Study Design

The present study involved 663 bachelor’s degree students at one of the public universities in Malaysia. This university is one of the leading research universities in Malaysia that has been recognised by government assessment. The respondents were determined by using multi-stage sampling, where seven faculties were randomly selected and one program from each selected faculty was randomly selected, namely, Bachelor of Forestry Science, Bachelor of Science (Nutrition and Community Health), Bachelor of Science in Biology with Honors, Bachelor of Science (Home Science), Bachelor of Engineering (Civil), Bachelor of Science (Food Studies) with Honors, and Bachelor of Science Biotechnology. Then, the randomly selected years (two) from each program were second-year and third-year students, and all students were invited to participate in this study. 

The duration of data gathering was between May 2019 and January 2020, through the platform “Google Form”. The leader of each selected course was approached and information was given concerning this study. Subsequently, with the help of the leader, the link was distributed to all the course mates to obtain access and complete the questionnaire. Prior to answering the questionnaire, respondents provided their informed consent, and a token of appreciation was given to those who completed the questionnaire. This study received endorsement from the Ethics Committee for Research Involving Human Subjects from the university (UPM/TNCPI/RMC/1.4.18.2).

### 2.2. Conceptual Framework

Mediator variable is the variable that explain the relationship between two variables (food security status and academic performance). The conceptual framework was developed to identify the potential mediator variables. The framework, which includes psychosocial factors as mediator (*M*), comprising of stress, anxiety, and depression, food security status as independent variable (*X*), and academic performance as dependent variable (*Y*). The relationships between X and M as well as M and Y have to be statistically significant in order to further analyse for potential mediator variables. The psychosocial factors were expected to potentially mediate the relationship between food security status and academic performance.

### 2.3. Measures

Data on food security status were measured using US Adult Food Security Survey Module (US-FSSM) [32]. This module is comprised of 10 items (see Appendix A) and for the affirmative responses of “often”, “sometimes”, “not true”, and “yes” is given one score. In contrast, non-affirmative answer is coded as zero score. The highest total score of ten indicates the highest severity of food insecurity. The respondents with a score of 2 or more indicate food insecurity. Cronbach’s alpha in this study is 0.80. 

Psychosocial factors were assessed by using Depression Anxiety Stress Scale-21 (DASS 21), which is a shortened version of the original DASS-42 that can examine all the psychometric properties to determine the level of SAD. It consists of 21-item self-report tool measuring attitudes and symptoms of SAD (see Appendix B). There are 21 items with 7 items each for stress, anxiety, and depression (SAD) [33]. For stress, the subscale emphasises insistent stimulation and pressure. Meanwhile, for anxiety, the items measure distress response and emotional stimulus, whereas depression assesses low mood, low self-esteem, and a lack of interest for the future [34]. Respondents were asked to score every item on a scale from zero (did not apply to me at all) to three (applied to me very much). Based on the DASS-42 manual, the sum scores were computed by adding up all the items per subscale and multiplying with 2. The total score for each subscale ranged between zero and 42, of which higher scores indicate greater levels of distress. It is noteworthy that DASS-21 is a screening tool that measures the severity of the symptoms and is not a diagnostic tool for mental health. This instrument is broadly recognized and practical due to its consistency, accessibility, and ease of use for the general population. The Cronbach’s alpha values in this study were relatively good: depression scale (α = 0.81), anxiety scale (α = 0.69), and stress scale (α = 0.73).

For academic performance, self-reported cumulative grade point average (CGPA) was classified into two groups according to the university grading system: first class honors (≥3.75) and honors (<3.75).

The information regarding demographic and socioeconomic characteristics were given (age, gender, living arrangement, marital status, parent’s educational background, parent’s occupation, and household income).

### 2.4. Data Analysis

All descriptive and correlative data were analysed by using IBM SPSS version 22 (IBM Corp, Armonk, NY, USA) [35]. Pearson correlation was used in order to analyse the relationship between each variable (food security status, stress, anxiety, depression, and academic performance) to include in mediation analysis. The statistical significance level was set at *p* < 0.05. Meanwhile mediation analysis was conducted using the PROCESS macro for SPSS [36] to examine the direct and indirect pathway through which food security status transmits its effect on academic performance through stress, anxiety, and depression. The PROCESS macro produces bias-corrected estimate calculated for 10,000 bootstrapped with 95% asymmetrical confidence intervals for the indirect effect. The relationship was found to be mediated if the indirect effects were significant and it exists when the lower and upper bounds for the 95% bias-corrected confidence intervals do not contain zero. Notably, Preacher and Hayes’ (2004) mediation method is a non-parametric, bootstrapping technique that proposes powerful and precise results for mediation analysis compared to previous traditional methods and is fully supported by Hair et al. (2014) [37]. 

## 3. Results

### 3.1. Characteristics of the Respondents

The total number of respondents involved in this study was 663. The mean age of the respondents was 21.9 years, and the majority were female (67.6%) (Table 1). Most of the respondents lived on campus (96.4%) and were single (96.2%). Over half the respondents had a working parent, with the father (82.5%) and mother (64.1%). Moreover, for the monthly household income, more than two-fifths of the respondents had an income of less than MYR 4850 (=USD 1164) (44.6%), and a smaller percentage of the respondents had a monthly household income of more than MYR 10,959 (=USD 2630.16). In addition, 62.8% were reported as food insecure and most respondents had a CGPA of less than 3.75 (88.6%), with a mean of 3.42 points. In terms of psychosocial factors, more than half of the respondents had normal levels of stress (65.5%), depression (59.9%), and anxiety (22.7%). 

### 3.2. Correlation Analysis

Pearson correlation was used to determine the correlation among the variables. All variables were correlated with one another except for relationship between food security status and stress (refer to Table 2). The relationship between stress and depression was the highest, *r* = 0.748, *p* < 0.01, while the relationship between food security status and academic performance was the lowest, *r* = −0.209, *p* < 0.01.

### 3.3. Mediation Analysis

To further analyse the psychosocial factors (anxiety and depression) on food security status and academic performance, the PROCESS macro for SPSS was used for mediation analysis. Stress variable is not included in this mediation analysis because the relationship between food security status and stress was not found to be significantly associated in the previous analysis (correlation analysis). In this study, two mediation models were proposed for mediator anxiety and depression. Firstly, this study focused on how anxiety influences the relationship between food insecurity and academic performance. The results showed that food insecurity was positively correlated to anxiety (β = 0.491, 95% CI: 0.0676, 0.6293, *p* < 0.05) and had a negative relationship with academic performance, (β = −0.0182, 95% CI: −0.0259, −0.0013, *p* < 0.001). When anxiety was a predictor, a significant and negative influence on academic performance was found (β = −0.0027, 95% CI: −0.0046, −0.0004, *p* < 0.05) (see Figure 1). The indirect effect was 0.0013 and was statistically not zero, as revealed by a 95% bootstrap confidence interval from −0.0022 to −0.0001. The direct effect between food insecurity and academic performance remained significant, *p* < 0.01, even after accounting for the indirect effect of anxiety, indicating partial mediation. As a result, those who experience food insecurity are more prone to experience anxiety (because 0.491 is positive), which in turn explains the decrease in academic performance (because 0.0027 is negative).

On the other hand, in the second mediation model, the results showed that food insecurity was positively correlated to depression (β = 0.3485, 95% CI: 0.2279, 0.7542, *p* < 0.05), and depression was negatively related to academic performance, (β = −0.0025, 95% CI: −0.0049, −0.0004, *p* < 0.05) (see Figure 2). Moreover, food insecurity had a negative relationship with academic performance, (β = −0.0186, 95% CI: −0.0255, −0.0108, *p* < 0.001). The significant indirect effect was 0.0009 and a 95% bootstrap confidence interval was revealed from −0.0146 to −0.0010 that was statistically below zero. The direct effect between food insecurity and academic performance remained significant, *p* < 0.01, even after accounting for the indirect effect of depression, indicating partial mediation. As a result of food insecurity, students are more exposed to having depression (because 0.3485 is positive), which in turn decreases students’ academic performance (because 0.0025 is negative).

## 4. Discussion

The current study revealed that approximately 62.8% of students had experienced food insecurity. These high prevalence data are in line with a recent systematic review on food security data in Malaysia, which reported the prevalence of food insecurity among university students in Malaysia is between 22% to 70% [9]. Moreover, this study reported the prevalence of students who are at risk of stress (34.5%), anxiety (77.3%), and depression (40.1%). The high prevalence of anxiety is very concerning because being anxious and nervous requires a lot of energy and these students will feel overly fatigued and restless. Therefore, the issue regarding food insecurity and psychosocial factors among university students should be considered as publicly alarming.

In the correlation analysis, this study found that significant correlations existed between food security status, anxiety, depression, and academic performance. Firstly, food security status and academic performance were negatively correlated. This is consistent with the past literature that indicated that food insecurity prevents a student’s success [7,11,38]. Morris disclosed that a higher proportion of food security students had a GPA of more than 3.0 compared to the low GPA range [11]. Meanwhile, students at the University of California reported a higher proportion of food secure students had an A average compared to students experiencing food insecurity [7]. Students who experienced food insecurity found it difficult to concentrate on studies due to insufficient money to obtain food [38]. The possible explanation is that, when they have no money to buy good quality food for energy needed by the body and brain, this can cause a decrease in their ability to concentrate in class.

As soon as the body does not acquire an adequate amount of food, the individual starts to become weary and can suffer from anxiety and lack of sleep. The body becomes fatigued and this disturbs the concentration ability in class. The student can feel exhausted because of their inadequate consumption of food causing energy depletion and worsening their ability to perform well academically [11]. Moreover, insufficient nutrients can cause a decrease in cognitive functioning [17]. The qualitative research of students in California describes how food insecurity impacts a student’s ability to concentrate on their studies [39]. Additionally, food insecurity may result in hunger feelings of weakness and discomfort caused by a lack of food. Being hungry impairs concentration, which can lead to poor academic performance [40].

Secondly, food insecurity affects students’ psychosocial factors, which are anxiety and depression, but no correlation was found between food security and stress. This finding shows consistency with the past literature [10,14,27]. Seemingly, food insecure students experienced higher odds of anxiety and depression. Emotional anguish results from the body’s inability to obtain enough nutritious food [41]. They become dissatisfied, which has an impact on their psychosocial well-being. Usually, food insecure students become concerned due to an inadequate amount of money to purchase food. Therefore, this results in emotional distress as they struggle to meet their basic needs through food. Subsequently, they consume less healthy food and skip meals due to their lack of affordability, which affects their mental health. Food insecurity among students has been linked to high levels of anxiety and depression, and it then acts as a stressor that interferes with their psychosocial health [8]. In this study, the effect of food insecurity on anxiety was higher compared to depression because more than half of the students in this study showed anxiety symptoms. Furthermore, anxiety is more common compared to depression as depression is more severe.

Thirdly, stress, anxiety, and depression are negatively associated with academic performance. When stress, anxiety, and depression increase, academic performance decreases. A similar previous study also found the association between high levels of stress [42], anxiety [16,43], and depression [16,44] and academic performance among students. The multi-functional role as a student exhausts their physical and mental health, and can interfere with their academic performance [45]. Several factors, such as decreased energy, low level of curiosity, difficulty making decisions, and difficulty concentrating, were found to be detrimental to academic performance as a result of poor mental health [45].

In addition, mediation analysis revealed that food security status also indirectly affects academic performance through psychosocial factors. The results revealed that anxiety and depression partially mediate the relationship between food security status and academic performance among students. Partially mediating means that there is still a relationship between food security status and academic life after the mediator was introduced. An assumption that can be made is that food insecure students perform poorly in their academic studies as a result of anxiety and depression. Though this study is the first local study to find a mechanism through which food insecurity affects academic performance, a few previous studies supported the identified individual pathway association [7,8] The researchers addressed the importance of consistent access to nutritionally adequate foods for students in higher education systems to ensure students’ wellness and success throughout their future career [7].

Since this study is still new and is only supported by two existing studies, our findings provide theoretical and practical contributions. The present study highlights a significant indirect pathway by which food insecurity is associated with poor academic performance. However, it is worth noting that the indirect pathway was analysed using a cross-sectional study design, which hinders causal explanations as well as understanding regarding chronological priority. Nonetheless, the findings do provide preliminary data suggestions for future study. Our findings propose that students with poor psychosocial health, such as high levels of anxiety and depression in the state of food insecurity, may be more likely to perform poorly in their academics. On the other hand, if students have good psychosocial health, such as a low level of anxiety and depression in the face of food insecurity, they may be able to maintain good academic performance. It is crucial for researchers to conduct experimental and longitudinal studies to help examine if and how psychosocial factors impact one’s experience of food insecurity in the prediction of academic performance. Therefore, it is important for the government and higher education institutions to revise the program and policy priorities for students at greater risk of food insecurity and conduct future interventions to test the potential pathway in order to know the effectiveness of the identified potential variables.

With the amplified observation that university food insecurity has gained in the literature, many universities have begun to take into account and address student food insecurity. There are several ways for food insecure students to access food. Campus food pantries, which secure, store, and distribute free food directly to students in need, and student financial literacy programmes have been the most common responses to student food insecurity [12]. Aside from those, meal plan donations are used to assist students in need, in which students receive a debit-like card and use it to purchase food on campus or other essential needs [46]. At an individual level, students need to know how to properly feed themselves with a good quality of food to attempt to reduce the negative impacts of being food insecure. Healthy food choices with essential macro and micro nutrients can help students avoid food insecurity issues as well as improve emotionally. On the other hand, several universities in the United States of America are opting for campus gardens, aimed at providing a sustainable food source to help people in need while promoting agricultural education to their students. In addition, students can enhance their leadership skills by participating in this programme, thus improving their social life and moving away from being stressed and lonely [10].

Although this study enhances the body of knowledge, several limitations must be addressed. Firstly, the findings could not be generalised and are representative of the setting, which only included one particular university, one study location that focused only on urbanised settings, and a study sample that included only a specific year of study. Aside from that, the data were gathered through respondents’ self-reports, which could lead to bias if respondents fail to answer honestly, particularly on sensitive questions. According to Bogner and Landrock [40], individuals may be influenced by social norms when they answer questions in the way researchers expect them to answer. Furthermore, respondents may prefer to respond for satisfaction rather than honesty. Despite that, the self-report questionnaire was the most practical data collection method and was found to provide more consistent and accurate data. Lastly, the present study only focused on psychosocial factors for potential mediators. (i.e., stress, anxiety, and depression). There may possibly be another potential variable that might become a mediator between food security status and academic performance. For example, Raskind et al. [41] found hope to be a potential mediator variable between food security status and academic performance among university students in Georgia.

## 5. Conclusions

In conclusion, the current study draws on current issues faced by university students, such as food insecurity, which is a significant issue for students, and this study discovered that it significantly impairs their well-being and academic achievement. At the same time, initiatives by student affairs and counselling units to give knowledge and expand students’ skills in managing their finances, stress management, and food literacy can critically yield benefits for students.

## Figures and Tables

**Figure 1 ijerph-19-05535-f001:**
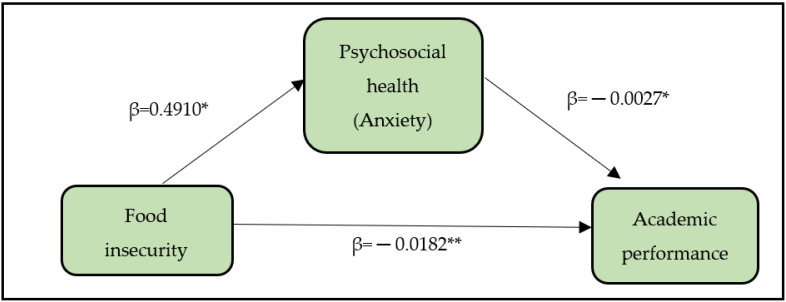
Framework for simple mediation (anxiety as mediator). * *p*-value < 0.05, ** *p*-value < 0.001.

**Figure 2 ijerph-19-05535-f002:**
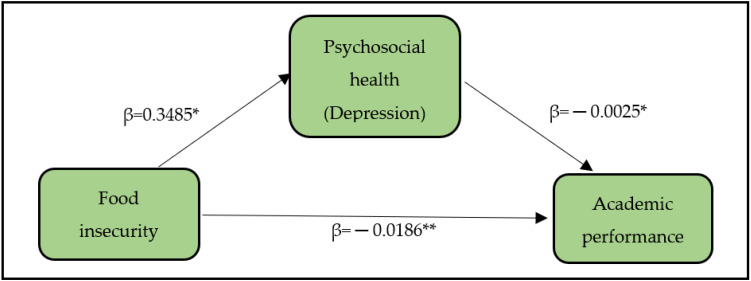
Framework for simple mediation (depression as mediator). * *p*-value < 0.05, ** *p*-value < 0.001.

**Table 1 ijerph-19-05535-t001:** Characteristics of respondents.

Variables	*n* (%)	Mean ± SD
Age (years)		21.98 ± 1.122
Gender		
Male	215 (32.4)	
Female	448 (67.6)	
Living arrangement		
On/Off Campus	639 (96.4)	
With family	24 (3.6)	
Marital Status		
Single	638 (96.2)	
Married	25 (3.8)	
Father occupation		
Working	415 (82.5)	
Not working	88 (17.5)	
Mother occupation		
Working	352 (64.1)	
Not working	180 (35.9)	
Monthly household income ^1^ (MYR *)		6746.65 (USD 1623.52) ± 5487.56
<MYR 4850 (USD 1164)	208 (44.6)	
MYR 4850—MYR 10,959 (USD 1164—USD 2630.16)	204 (43.8)	
>MYR 10,959 (USD 2630.16)	54 (11.6)	
Food security status		
Food secure	244 (37.2)	
Food insecure	412 (62.8)	
CGPA ^2^		3.42 ± 0.208
<3.75	534 (88.6)	
≥3.75	69 (11.4)	
Stress		25.50 ± 15.178
Normal	434 (65.5)	
At risk	229 (34.5)	
Anxiety		25.82 ± 14.623
Normal	150 (22.7)	
At risk	510 (77.3)	
Depression		17.79 ± 15.494
Normal	396 (59.9)	
At risk	265 (40.1)	

^1^ Income based on thresholds of monthly household gross income Malaysia 2019. * MYR 1 (=USD 0.24). ^2^ Cumulative grade point average (CGPA).

**Table 2 ijerph-19-05535-t002:** Correlation between food security status associated with stress, anxiety, and depression.

	1	2	3	4	5
Food security status		−0.209 **	0.061	0.141 **	0.083 *
2. Academic performance			−0.117 **	−0.122 *	−0.112 **
3. Stress				0.724 **	0.748 **
4. Anxiety					0.603 **
5. Depression					

* Significant at *p* < 0.05. ** Significant at *p* < 0.001.

## Data Availability

The data presented in this study are available on request from the corresponding author. The data are not publicly available due to privacy and ethical restrictions.

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
