# Peer review of "Psychosocial Factors as Mediator to Food Security Status and Academic Performance among University Students"

_ijerph, 2022, doi:10.3390/ijerph19095535_

Round 1

Reviewer 1 Report

This study presents the results of investigation on the role of psychosocial factors in mediating the relationship between food security and academic performance of university students. Generally, this manuscript needs substantial improvements. What concern me the most are the methodology section, results, and discussion. I have following comments.  

Page 2, line 51. It is indicated that most studies on food insecurity in university/college students found that the results were not on the good side. What do you mean by “not on the good side”. Please provide a concrete explanation and insert citations.

Page 2, line 71. Is food security the economic factor?

Page 2, line 76, 77. Please explain in detail on how students’ insufficient nutrition negatively affects students’ concentration and cognitive function.

The discussions on the relationship between psychosocial factors (stress, anxiety and depression) and academic performance are not clear. Please explain how those psychosocial factors possibly affect students’ academic performance.

In reality, many students with food insecurity could also achieve high academic performance. Therefore, students’ academic performance is not entirely affected by  psychosocial factors which are assumed to be affected by food security. This exception should be discussed as well. It is possible that food insecurity could positively affect students’ academic performance through motivation to achieve food security in the future.

Entire references are very old. More recent research works should be included.

Section 1.3, please clearly express the significance of this exploration. Please clearly explain how knowing the interrelation between food security status, psychosocial factors, and academic performance is important.

Please add the section of conceptual framework, and explain possible relationships among each variable based on theoretical perspectives. It would be more appropriate to explain how each psychosocial variable could mediate the effect of food insecurity on academic performance.

I am curious why students’ academic performance was measured by using only 2 scales, equal or higher than 3.75 and less than 3.75. In the data analysis, how did you put this type of data in the analysis? Was it treated as continuous variable of discontinuous one?

Regarding methodological section, please add the questions used for measuring each variable. Particularly, I wonder how food security was measured.

In the section 3.3. (Mediation analysis), I wonder why the mediating effect of stress variable was not analyzed and shown. Though the result was not statistically significant. The result should be presented.

Figure 1 and 2 were not in a good format. In addition, it should be noted that a higher of food security variable means “food insecurity”. Another option, in figure 1 and 2, please replace the term “food security” with “food insecurity”.

Please also indicate which psychosocial factor had the most mediating effect on the relationship between food insecurity and academic performance.

Page 8, line 12. Students who experienced food insecurity found it difficult to concentrate on studies due to insufficient money to get food [43]. I disagree with this statement. It seems like students who are having less money to get food would be hard to achieve a good academic performance. Really?

 Students can actually access food from several ways. They may bring food from their home. In addition, through they have less money, they may know how to spend those money for buying good quality of food. I suggest that It is important to discuss about quantity and quality of food that students consume as well.

In the discussion, please explain why the effect of food insecurity on anxiety was higher than the effect of food insecurity on depression?

Page 9, line 367, What do you mean by “poor academic performance”?

Please add concrete practical contributions of this research in the discussion.

Reviewer 2 Report

This is a current and relevant topic. I find the current study to be interesting and an important contribution to the literature. The following list of observations is meant to assist you with revision to this work. I appreciate the dedication to this research.

The nature of this study being conducted in Malaysia is very interesting. Please consider building up a case of Malaysian universities and thus students. 

In the design, consider further explaining the sampling. Why all science students? Why no first year or senior students? The rationale provided is not scientifically/ data driven as much as it is a justification of convenience.

Provide some rationale for the demographic information collected and the later discuss it. 

Table 2 can use more set-up in the narrative and then more interpretation and application, perhaps in the discussion.

Specific critique.

  • Line 259-260 - first mention of indirect affect of anxiety.
  • Line 301 has a typo.
  • Line 322-324, not sure where that statement came from.
  • Line 341- 342, where does mediating came from? Further describe.
  • 347-348, seems like some context is necessary to compare universities in different countries, different SES, etc. Why are they comparable - please establish the scope of comparison. 358, comparing again with colleges in California, how do we relate this universities?
  • 388, the line seems out of place and not really relevant but the paper 
  • Lines 405-407, citing an article that studied children 4-6 in Canada, are not sure if they can relate their findings to this article because of the population and location. 
  • They mention their own study once, but they don't discuss much of their results.
  • The discussion does not relate the material back to the study university in a manner that adds to the understanding of the findings and future direction.
  • The discussion and conclusion felt repetitive. Please add direction for the readers.

Round 2

Reviewer 1 Report

Authors have addressed all of my comments. I am satisfied with the revised version.